# A Tension-Band Wiring Technique for Direct Fixation of a Chaput Tubercle Fracture: Technical Note

**DOI:** 10.3390/medicina58081005

**Published:** 2022-07-27

**Authors:** Eui-Dong Yeo, Ki-Jin Jung, Yong-Cheol Hong, Chang-Hwa Hong, Hong-Seop Lee, Sung-Hun Won, Sung-Joon Yoon, Sung-Hwan Kim, Jae-Young Ji, Dhong-Won Lee, Woo-Jong Kim

**Affiliations:** 1Veterans Health Service Medical Center, Department of Orthopaedic Surgery, Seoul 05368, Korea; angel_doctor@naver.com; 2Department of Orthopaedic Surgery, Soonchunhyang University Hospital Cheonan, 31, Suncheonhyang 6-gil, Dongam-gu, Cheonan 31151, Korea; c89546@schmc.ac.kr (K.-J.J.); ryanhong90@gmail.com (Y.-C.H.); chhong@schmc.ac.kr (C.-H.H.); yunsj0103@naver.com (S.-J.Y.); 3Nowon Eulji Medical Center, Department of Foot and Ankle Surgery, Eulji University, 68, Hangeulbiseok-ro, Nowon-gu, Seoul 01830, Korea; sup4036@naver.com; 4Department of Orthopaedic Surgery, Soonchunhyang University Hospital Seoul, 59, Daesagwan-ro, Yongsan-gu, Seoul 04401, Korea; orthowon@schmc.ac.kr; 5Department of Orthopaedic Surgery, Soonchunhyang University Hospital Bucheon, 170, Jomaru-ro, Wonmi-gu, Gyeonggi-do, Bucheon-si 14584, Korea; sjk9528@naver.com; 6Department of Anesthesiology and Pain Medicine, Soonchunhyang University Hospital Cheonan, 31, Suncheonhyang 6-gil, Dongam-gu, Cheonan 31151, Korea; phmjjy@naver.com; 7Konkuk University Medical Center, Department of Orthopaedic Surgery, 120-1, Neungdong-ro, Gwangjin-gu, Seoul 05030, Korea; bestal@naver.com

**Keywords:** ankle fracture, syndesmosis injury, Chaput tubercle, avulsion fracture, tension-band wiring

## Abstract

Few reports have described direct fixation of the Chaput tubercle; screw fixation is usually employed. Herein, we introduce a novel technique for Chaput tubercle fixation using tension-band wiring. This technique is applicable to fractured tubercles of various sizes and has the advantage that the fragment breakage that may occur during screw fixation is impossible. In addition, our technique increases fixation strength.

## 1. Introduction

Distal tibiofibular syndesmotic injury associated with ankle fracture accounts for 10% of all ankle fractures, and up to 20% are treated surgically as rotational ankle fractures [1,2,3]. Distal tibiofibular syndesmosis is critical for maintenance of ankle congruency and integrity during weight-bearing; unstable syndesmosis requires surgery [4,5]. Anatomical reduction of an ankle fracture with stabilization of any accompanying syndesmosis injury is essential to ensure good, long-term functional results and to prevent post-traumatic arthritis [1,6]. The anterior inferior tibiofibular ligament (AITFL) is the strongest of the four ligaments of the syndesmosis and plays a prime role in stability; it prevents displacement of the distal fibula outward from the mortis when an external rotation force is applied to the ankle joint [7,8]. The AITFL is attached to the anterior tibial tubercle on the distal tibial side (this tubercle is better known as the “Chaput tubercle”). Thus, a fracture of the tubercle is generally termed a “Chaput fracture”, reflecting indirect injury of the syndesmosis [9]. Recent studies have suggested that direct fixation of fracture fragments is optimal for treating syndesmosis joint instability caused by a Chaput fracture [10,11]. Most direct fixation methods employ K-wires or screws [10,11,12], but this is impossible when the fracture fragments are small; in addition, the fixation strength is weaker than that of tension-band wiring (TBW) [13]. Here, we present a novel tension-band wiring technique that handles fracture fragments of various sizes and increases fixation strength. Such wiring is commonly performed in operating rooms.

## 2. Surgical Technique

This technical note was approved by the Institutional Review Board of Soonchunhyang University Cheonan Hospital (approval no. 2022–06–037, 2022-06-22). The patient provided written informed consent for the publication of this report and the accompanying images.

The procedure is performed under general or spinal anesthesia, or a lower extremity nerve block. The patient is placed supine, and the lower extremity is prepared and draped in the usual sterile manner. A tourniquet is inflated to ensure a bloodless surgical field. If a fibular fracture is also present, a curved anterolateral approach is chosen (Figure 1, dotted line). Through this line, first, reduction of the fibular fracture and, in most cases, plate fixation, and then approach to Chaput fragment are attempted. In the absence of a fibular fracture, a small anterolateral incision (2–3 cm) is created (solid line) directly over the palpable Chaput tubercle of the distal tibia. The anterolateral tibial fragment (the Chaput fracture) is identified and the fracture is cleared of debris. The fracture is reduced and temporarily fixed using small point-reduction forceps. The extent of reduction and the congruency of the articular surface are confirmed via intraoperative fluoroscopy. This also serves to ensure that the hardware is appropriately positioned and that the articular surface is not displaced. Then, two 1.2–1.6-mm (the diameter varies by the size of the fracture fragment) Kirschner (K)-wires are inserted proximally from the end edge of the Chaput fragment through the fracture site. These wires prevent fracture rotation and are later used to anchor a figure-of-eight wire distally. To ensure that the K-wires are fully seated on the end of the tubercle after the ends have been bent, they are pulled back slightly. Next, the medial incision site over the distal tibia is retracted to expose the anterolateral tibial border approximately 2 to 3 cm cephalad to the fracture site, and a ϕ 4.0-mm, cancellous, full-thread screw is inserted without tapping and without complete seating (Figure 2). Stainless-steel wires (ϕ 0.8 mm) are looped around the screw and the K-wires in a figure-of-eight manner (Figure 3). Then, the loops are tightened to ensure that they cling to the anteroinferior surface of the distal Chaput fragment and the steel wires are twisted at the points of insertion in the K-wires. Next, the two K-wires are cut obliquely, bent medially, and tapped into the medial malleolus; they are now fully seated. If the fracture fragment is small, thinner K-wires and steel wires are used. If a fracture fragment is impacted, it is possible to first attempt a bone graft. Figure 4 shows a postoperative plain X-ray of open reduction/internal fixation of a Chaput fracture using this technique (Figure 4). Axial computed tomography confirmed that both the reduction and the fixation were satisfactory (Figure 5). For this patient, a short leg splint was prescribed postoperatively for about 1–2 weeks. Then, the cast or range-of-motion (ROM) ankle walker brace was changed and the patient was instructed not to place weight on the limb for a further 4 weeks. ROM exercise commenced at 4 weeks after surgery, and then weight-bearing was gradually restored using the ROM ankle walker brace. After 6 weeks, full weight-bearing commenced, and the brace was removed at 8 weeks. A clinical union was confirmed 6 weeks after surgery and a radiologic union was confirmed on follow-up CT 3 months after surgery. The patient demographics and clinical analysis results are presented in Table 1.

## 3. Discussion

Several studies have reported the prognoses of Chaput fracture treatment. Haraguchi et al. [14] reported that the union rate of non-operated Chaput tubercle fractures was only 65%. Birnie et al. [15] reported that four patients (6.2%) of an AITL avulsion fracture group required additional surgery. Zhao et al. [16] performed open reduction/internal fixation on 15 adult patients with ankle fractures involving Tillaux–Chaput fractures. The mean AOFAS score was 87, with an excellent or good rate of 80%: excellent in nine cases, good in three, and fair in three. Bae et al. [11] performed direct avulsion fracture fixation on patients evidencing syndesmotic instability after malleolar fractures combined with AITL avulsion fractures. Syndesmotic stability was achieved by 45 (83.3%) of 54 patients; the remaining 9 (16.7%) required additional syndesmosis screw fixation.

Direct fixation of a fractured Chaput tubercle ensures not only bone-to-bone fixation of the anterior syndesmosis but also correct positioning of the fibula into the tibial incisura [17]. A few studies have found that inadequately treated bony avulsions of the tibiofibular syndesmosis can trigger translational or rotational malposition, which damages the structure of the ankle mortise [18,19]. After such a postoperative event, revision surgery should be urgently performed.

Several methods for direct fixation of Chaput fractures have been described. Chung et al. [10] reported good results after direct fixation of anteroinferior, tibiofibular, ligament avulsion fractures using K-wires, mini-screws, or absorbable suture materials. Six cases presented with Chaput fractures, including four of the modified Wagstaffe classification type III and two of type IV. However, the fixation materials were not described. Rammelt et al. [20] used plates, screws, and suture anchors. Gasparova et al. [12] found that screw fixation was optimal for monofragmented fractures, but plate fixation was best for multifragmentary fractures.

Historically, TBW has been recommended for AO patients when a fragment is too small for screw fixation into an avulsion fracture or when screw fixation is inadequate, such as in osteoporotic bone [21]. However, TBW has gradually become used to fix large fragments; it is increasingly recognized that TBW ensures good fusion rates and good functional results [22]. We reviewed the literature when applying TBW to treat Chaput fractures.

However, a limitation of this study is that the fixation strength of this technique was not compared with other devices in fixing the Chaput tubercle fragment. It is thought that cadaver studies for strength comparison are necessary.

Our technique is independent of the size of the fractured Chaput tubercle. Neither high-level surgical skill nor extensive experience are required.

## 4. Conclusions

This technique is applicable to fractured tubercles of various sizes. Additionally, it is advantageous when there is a possibility that fragment breakage may occur during other device fixation.

## Figures and Tables

**Figure 1 medicina-58-01005-f001:**
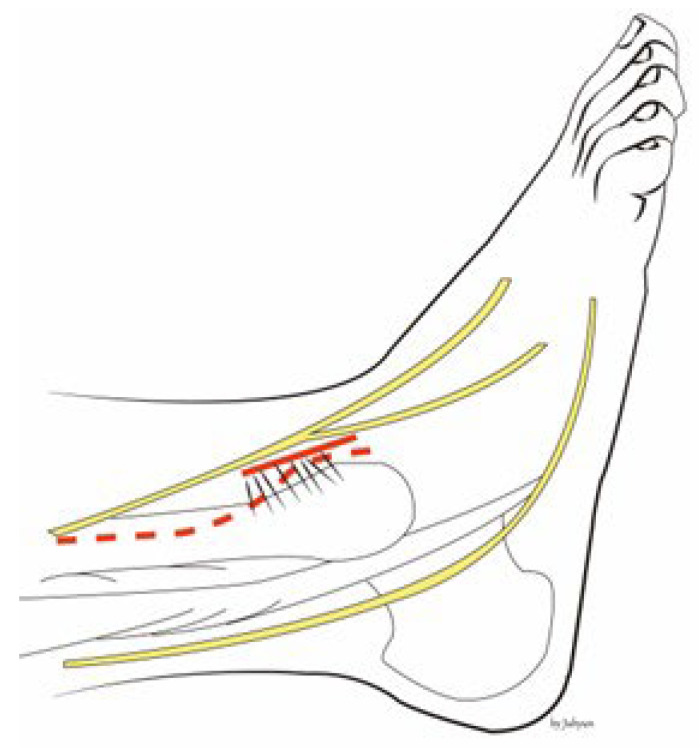
If a fibular fracture is also present, we take a curved anterolateral approach (dotted line). In the absence of such a fracture, a small anterolateral incision (about 2–3 cm) is created (solid line) directly over the palpable Chaput tubercle of the distal tibia.

**Figure 2 medicina-58-01005-f002:**
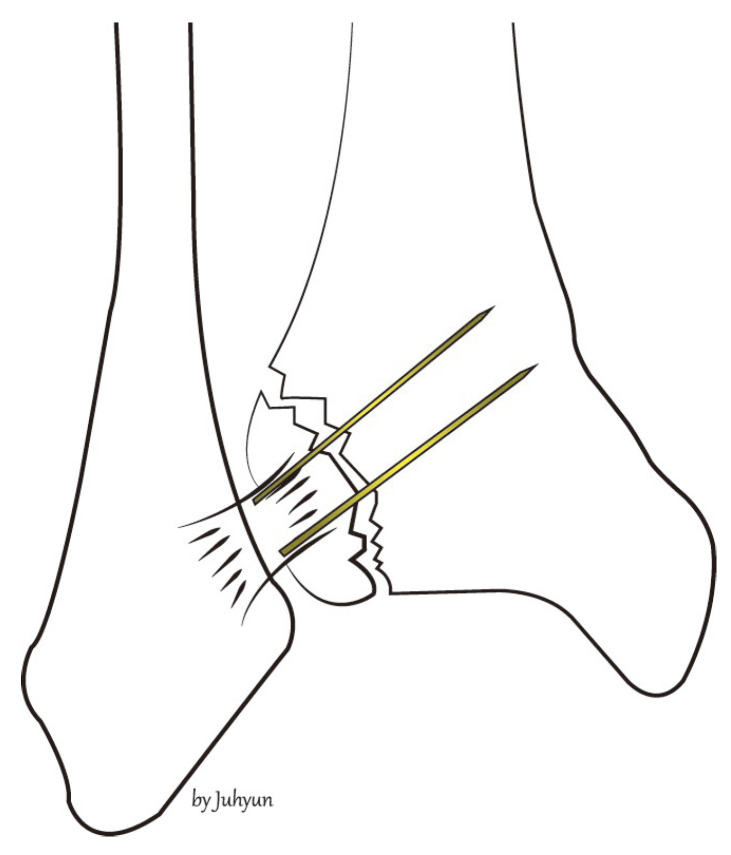
After the Chaput fracture has been reduced, two K-wires and screws are fixed.

**Figure 3 medicina-58-01005-f003:**
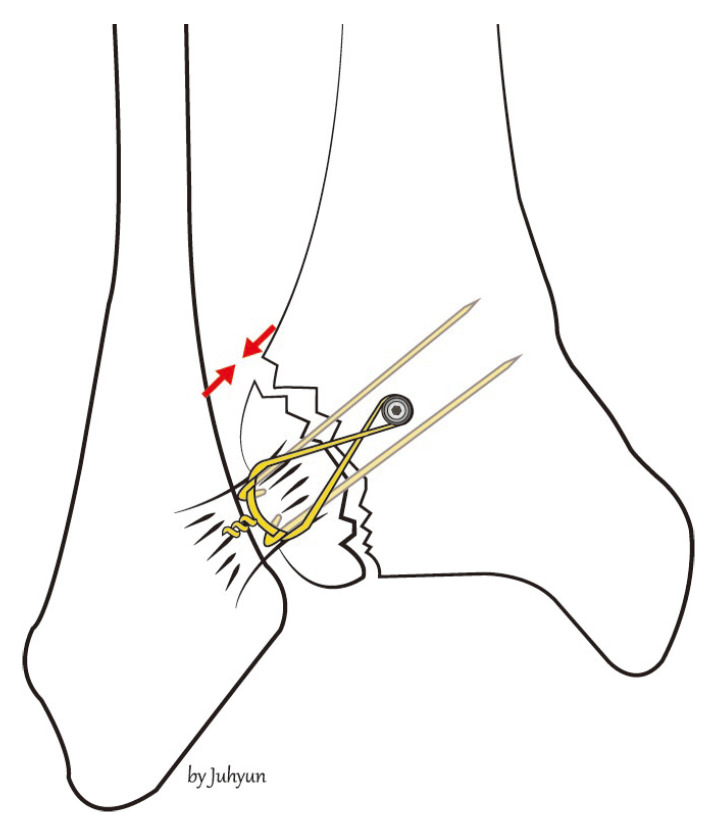
Stainless-steel wires are looped around the screw and the K-wires in a figure-of-eight manner.

**Figure 4 medicina-58-01005-f004:**
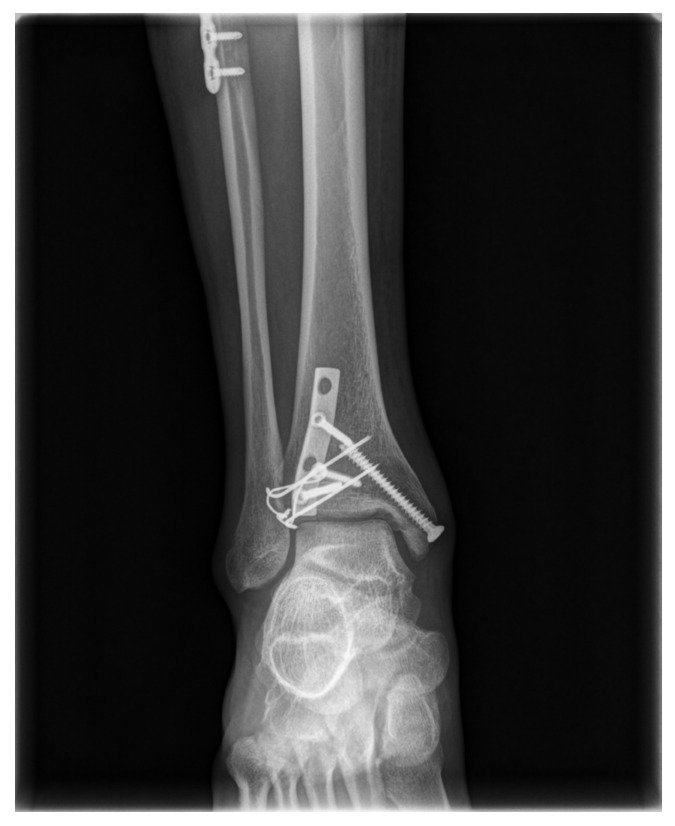
A postoperative, plain anteroposterior radiograph shows a Chaput fracture fixed using the new technique.

**Figure 5 medicina-58-01005-f005:**
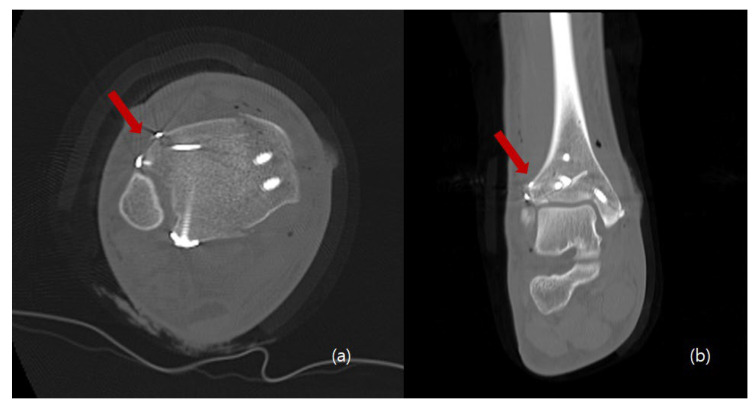
Postoperative axial (**a**) and coronal (**b**) computed tomography images show that the Chaput fracture exhibited good reduction, compression, and fixation (arrow).

**Table 1 medicina-58-01005-t001:** Patient demographics and results.

Pt. No.	Age	Sex	Cause	Lauge-Hansen Classification	Injury to Surgery Interval (hr)	* Procedure Time (min)	Injured Side	Follow-Up (mo)	OMAS		VAS Score		Interval to Union (wk)	Complications
									Pre	Post	Pre	Post		
1	57	F	S	SER IV	138	19	Left	9	30	80	8	1	14	None
2	76	F	S	SER IV	118	20	Left	14	25	85	7	0	15	None
3	56	F	S	SER II	98	18	Right	6	30	90	8	0	14	None
4	39	M	S	SER IV	87	18	Left	13	35	95	8	0	12	None
5	58	F	S	SER IV	282	19	Left	12	0	80	9	1	14	None
6	79	F	TA	PER IV	97	17	Left	6	0	70	8	0	16	None
7	16	M	TA	PER IV	68	18	Right	4	40	95	8	0	13	None
8	17	M	TA	PER II	39	18	Right	4	30	90	7	0	12	None
9	53	F	S	SER IV	258	16	Left	3	0	60	9	0	14	None
Mean	50.1	NA	NA	NA	131.7	18.1	NA	7.9	21.1	82.8	8	0.2	13.8	NA
SD	22.5	NA	NA	NA	83.5	1.2	NA	4.2	164	11.8	0.7	0.4	1.3	NA
*p-*value										0.007		0.006		

Abbreviations: Pt. No., patient number; OMAS, Olerud–Molander Ankle Score; VAS, visual analog scale; Pre, preoperative; Post, postoperative; F, female; M, male; S, slip down; TA, traffic accident; SER, supination external rotation; NA, not applicable; SD, standard deviation; * procedure time, tension-band wiring time. Statistical analysis was performed by a statistical expert. All calculations were made using SPSS, version 26.0, software (IBM Corp., Armonk, NY, USA). Quantitative variables are expressed as the mean ± standard deviation. The pre- and postoperative VAS and OMAS scores were compared using the Wilcoxon signed-rank test. A two-sided test with *p* < 0.05 was considered statistically significant.

## Data Availability

Data sharing is not applicable to this article as no datasets were generated or analyzed during the current study.

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
