# Peer review of "A Tension-Band Wiring Technique for Direct Fixation of a Chaput Tubercle Fracture: Technical Note"

_medicina, 2022, doi:10.3390/medicina58081005_

Round 1

Reviewer 1 Report

Dear Authors:

         Thank You for sharing Your work with the scientific community. TBW is a well-known methods of fixation and it was initially conceived to compress fracture sites dinamically, with forces applied with range of motion. After that, it was realized TBW could be used in small fragments that were at risk to be smashed if screws were used. The distal tibia is a site were such small fragments have been fixed with TBW for a long time. The novelty of this paper is in the segment fixed.

       Anyway, in my opinion, some limitations are present into the paper, and they should be addressed before publication. In 2022, if a novel surgical technique is reported, at least a case series should be included: can You implement the paper with results? The sentence "a high fixation force is guaranteed" (line 136, and abstract) should be removed or You should cite the paper that pointed it out, as You did not do any mechanical study in it. Also, English should be somewhere revised.

Author Response

Reviewer 1

- First of all, thank you very much for reviewing my insufficient paper despite your busy schedule.

Anyway, in my opinion, some limitations are present into the paper, and they should be addressed before publication.

In 2022, if a novel surgical technique is reported, at least a case series should be included: can You implement the paper with results?

: Yes, I am preparing a case series on chaput tubercle fracture as a sequel to this. This paper is a technical note and summarizes information about a brief case series in table format. Next, if you allow me, I will prepare a case series as a sequel.

The sentence "a high fixation force is guaranteed" (line 136, and abstract) should be removed or You should cite the paper that pointed it out, as You did not do any mechanical study in it. Also, English should be somewhere revised. :

: I agree with your point. I'll delete this sentence "a high fixation force is guaranteed". As reviewers 1 and 3 also pointed out, a limitation that a mechanical study is necessary was added to the last part of the discussion. thank you

Reviewer 2 Report

Interesting technique and presented well. Unlike to change practice but a good read.

Author Response

Thank you for your interest in my paper.

Reviewer 3 Report

The authors

I congratulation your work in devicing a surgical technique for the fixation of chaput fracture

I have some queries that needs clarification before recommending your article for publication

Having not compared the strength with screw fixation you cannot conclude the superiority of the current technique

Since both the wires are put in the anterior aspect of the bone surface did the authors see any posterior opening in any of the their cases?

Any implant related issues noted with the current technique?

Limitations of the technique?

How many cases did the authors use this technique?

Author Response

Reviewer 3

Having not compared the strength with screw fixation you cannot conclude the superiority of the current technique

: Your point is correct. Therefore, we are planning to study the carcass of screw and TBW. The expression of superiority actually meant that the selection range was better depending on the size of the bone fragment, but it seems that the expression was poor.

At the end of the discussion, as a limitation of this study, I will explain that a comparative study of the strength of screw and TBW Chaput tubercle fragment is necessary to avoid misunderstandings by readers.

Since both the wires are put in the anterior aspect of the bone surface did the authors see any posterior opening in any of their cases?

: In fact, fixing the far cortex has a greater holding force, but the authors did not fix the far cortex.

In all cases there was no problem at all. thank you

Any implant related issues noted with the current technique?

: Yes, in fact, there were concerns about skin irritation or wound problems caused by implants when trying this technology at the beginning. But in all cases, there were no problems at all. thank you

Limitations of the technique?

: The technique itself has no special limitations. However, I think that comparison with other fixing materials pointed out earlier is a limitation. This point has been described in the discussion. thank you

How many cases did the authors use this technique?

: I have operated more than 9 cases, and I am preparing a “case series” following this paper. I would like to submit it to “medicina” as soon as I am ready. thank you

Reviewer 4 Report

This manuscript aims to present a technical note on a novel technique for Chaput tubercle fixation using tension-band wiring

1. The status of fracture union has not been commented upon in the manuscript

2. Figure number 4 shows that a plate has been applied on the anterolateral aspect of distal tibia. No mention regarding the same has been made in the surgical technique section.

3. Please include details of the cases treated by this technique, in tabular format, along with union status, time taken for the same and functional outcome on final follow up.

4. A claim has been made-'the fragment breakage that may occur during screw fixation is impossible', which does not appear to be completely true always. The language needs to be toned down.

Author Response

Reviewer 4

This manuscript aims to present a technical note on a novel technique for Chaput tubercle fixation using tension-band wiring

  1. The status of fracture union has not been commented upon in the manuscript

Yes, I agree with what you pointed out. At the end of the surgical technical, union was further explained. For reference, I checked the patients I operated on at the outpatient clinic for a set period, and there was little difference in the union period between the patients. thank you

  1. Figure number 4 shows that a plate has been applied on the anterolateral aspect of distal tibia. No mention regarding the same has been made in the surgical technique section.

 : Yes, I agree with what you pointed out. In surgical technique, we have added what you pointed out to the incision line in figure 1. thank you

  1. Please include details of the cases treated by this technique, in tabular format, along with union status, time taken for the same and functional outcome on final follow up.

 : Yes, I have attached it as you pointed out.

  1. A claim has been made-'the fragment breakage that may occur during screw fixation is impossible', which does not appear to be completely true always. The language needs to be toned down.

: Yes, I agree with what you pointed out. Corrected the expression to "This technique is applicable to fractured tubercles of various sizes and has the advantage that when there is a possibility that fragment breakage may occur during other device fixation".

Round 2

Reviewer 3 Report

Dear authors 

The concluding statement is too complex I suggest making it simple to make it more understandable to the readers. 

In limitation you need not say there were no studies instead i suggest you admitting that a comparative study was not done to comment of the comparative fixation strength of the current technique 

Author Response

Reviewer 3

The concluding statement is too complex I suggest making it simple to make it more understandable to the readers.

: I agree with you. However, in order to highlight some of the advantages of this technique, the sentence "size" and "breakage" of the fracture fragment have been modified to be simpler. thank you,

In limitation you need not say there were no studies instead i suggest you admitting that a comparative study was not done to comment of the comparative fixation strength of the current technique

: I agree with your opinion. Therefore, the limitation has been corrected as pointed out. thank you.
